# The Identification and Function of Linc01615 on Influenza Virus Infection and Antiviral Response

**DOI:** 10.3390/ijms25126584

**Published:** 2024-06-14

**Authors:** Guihu Yin, Jianing Hu, Xiangyu Huang, Yiqin Cai, Zichen Gao, Xinyu Guo, Xiuli Feng

**Affiliations:** 1Key Laboratory of Animal Microbiology of China’s Ministry of Agriculture, College of Veterinary Medicine, Nanjing Agricultural University, Nanjing 210095, China; yinguihu@stu.njau.edu.cn (G.Y.); 2021207047@stu.njau.edu.cn (J.H.); 2020107044@stu.njau.edu.cn (X.H.); 2021107050@stu.njau.edu.cn (Y.C.); 2022107058@stu.njau.edu.cn (Z.G.); 2022807117@stu.njau.edu.cn (X.G.); 2MOE Joint International Research Laboratory of Animal Health and Food Safety, College of Veterinary Medicine, Nanjing Agricultural University, Nanjing 210095, China

**Keywords:** DHX9, IAV, RNA-seq, Linc01615, CLIP-seq

## Abstract

Influenza virus infection poses a great threat to human health globally each year. Non-coding RNAs (ncRNAs) in the human genome have been reported to participate in the replication process of the influenza virus, among which there are still many unknowns about Long Intergenic Non-Coding RNAs (LincRNAs) in the cell cycle of viral infections. Here, we observed an increased expression of Linc01615 in A549 cells upon influenza virus PR8 infection, accompanied by the successful activation of the intracellular immune system. The knockdown of Linc01615 using the shRNAs promoted the proliferation of the influenza A virus, and the intracellular immune system was inhibited, in which the expressions of IFN-β, IL-28A, IL-29, ISG-15, MX1, and MX2 were decreased. Predictions from the catRAPID website suggested a potential interaction between Linc01615 and DHX9. Also, knocking down Linc01615 promoted influenza virus proliferation. The subsequent transcriptome sequencing results indicated a decrease in Linc01615 expression after influenza virus infection when DHX9 was knocked down. Further analysis through cross-linking immunoprecipitation and high-throughput sequencing (CLIP-seq) in HEK293 cells stably expressing DHX9 confirmed the interaction between DHX9 and Linc01615. We speculate that DHX9 may interact with Linc01615 to partake in influenza virus replication and that Linc01615 helps to activate the intracellular immune system. These findings suggest a deeper connection between DHX9 and Linc01615, which highlights the significant role of Linc01615 in the influenza virus replication process. This research provides valuable insights into understanding influenza virus replication and offers new targets for preventing influenza virus infections.

## 1. Introduction

Influenza is an acute infectious respiratory disease, and influenza virus infection breaks out at different scales around the world every year with the season. It is estimated that, globally, influenza causes about 3 million to 5 million serious cases and about 290,000 to 650,000 deaths related to respiratory diseases every year [1]. Influenza viruses belong to the Orthomyxoviridae family and are negative-sense RNA viruses. The types A and B influenza viruses are highly pathogenic and collectively cause significant economic losses worldwide each year. The type C influenza virus is less prevalent and typically causes mild infections, while it is unknown whether the type D influenza virus can infect humans [2]. The recurring key process in these epidemics is the evolution of the virus to evade immunity induced by previous infections or vaccination [3]. The influenza virus is an enveloped, segmented, and single-stranded negative-sense RNA virus [4]. The segmented nature of the influenza virus genome facilitates the emergence of pandemic strains. The coinfection of two viruses into a single cell allows the segment reassortment between the different subtype viruses, including a process known as antigenic shift, which can generate novel antigenic viruses capable of spreading and causing disease in humans [5,6].

These recurrent key processes in epidemics involve the evolution of viruses to evade immunity generated by previous infections or vaccinations. Currently, influenza virus vaccines serve as effective countermeasures against infection. However, due to antigenic drift, they require almost annual reformulation. Additionally, these vaccines cannot preempt novel pandemic strains. The timely production of pandemic vaccines remains problematic due to limitations in current technologies [7]. On the other hand, since the outbreak of Severe Acute Respiratory Syndrome Coronavirus 2 (SARS-CoV-2), the virus has spread to over 100 countries, resulting in more than 5 million deaths [8]. Over time, the viral genome has undergone changes that enable it to evade vaccine-induced immunity [9]. In severe cases, SARS-CoV-2 can trigger a “cytokine storm” a hyperinflammatory condition that leads to increased pain intensity and prolonged pain duration [10]. In addition, there is evidence suggesting that co-infection with the type A influenza virus enhances the transmissibility of SARS-CoV-2 [11]. There is an urgent need for in-depth research into the lifecycle of influenza viruses. A thorough understanding of the influenza virus lifecycle has the potential to offer new insights into preventive strategies.

Additionally, the viral genome possesses replicative capabilities, but due to its small size, many viruses have evolved to coordinate with cellular machinery to provide necessary alternative functions. For instance, most positive-sense and double-stranded RNA viruses have open reading frames (ORFs) encoding RNA helicases, while the genomes of retroviruses lack virally encoded helicases [12]. Similarly, the influenza virus relies on host factors to facilitate each step of viral replication and hijack the host translation machinery by which influenza virus replication and translation can proceed normally [13,14,15]. The Human Genome Projects ENCODE and GENCODE have revealed that over 85% of the human genome can be transcribed into RNA, with only 3% encoding proteins. Non-coding RNAs (ncRNAs) constitute 60–70% of RNA, indicating that a substantial portion of the genome is non-protein-coding [16,17,18]. Non-coding RNAs are categorized into small ncRNAs and long ncRNAs (lncRNAs) based on length [19]. Long non-coding RNAs (lncRNAs) are defined as transcripts longer than 500 nucleotides that do not encode proteins. Recently, it has become apparent that lncRNAs represent a broad definition encompassing various types of transcripts with no obvious protein-coding potential, and some lncRNAs defined in this way have been shown to encode micropeptides [20,21]. The annotated number of lncRNAs has been steadily increasing, currently exceeding 20,000, with the majority lacking known functions [22]. However, a relatively small number of lncRNAs are associated with a wide range of biological processes through different molecular mechanisms. Long intergenic non-coding RNAs (LincRNAs) are a subset of lncRNAs that do not overlap with protein-coding genes. Some researchers argue that categorizing LincRNAs as coding RNAs may be more appropriate because some genes annotated as LincRNAs include small open reading frames (smORFs) [23]. LincRNA genes possess various features that are distinct from mRNA-coding genes and play roles in reshaping the chromatin and genome structures, RNA stability, and transcriptional regulation (including enhancer-associated activities) [23]. The annotation and discovery of LincRNAs are rapidly advancing, contributing to an evolving understanding of the functional landscape of the non-coding-RNA world. Increasing evidence suggests that lncRNAs play crucial roles in transcriptional and post-transcriptional regulation of gene expression across various biological processes, including X chromosome inactivation (Xist/Tsix) [24,25], genomic imprinting (H19 and Air) [26,27], stem cell pluripotency [28], development [29], cancer metastasis (HOTAIR) [30], and atherosclerosis (Anril) [31]. The innate immune system serves as the first line of defense against microbial pathogens [32]. LncRNAs have also begun to functionally contribute to controlling gene expression in the immune system. Recent studies collectively indicate that lncRNAs play important functional roles in innate immune cells, such as phagocytes. Notably, the discovery of LincRNA-Cox2 [33], Lethe [34], PACER [35], and THRIL [36] (lncRNA regulating TNFα interaction with hnRNPL) has once again highlighted exciting functions of lncRNAs associated with the control of gene expression in immune cells. LncRNAs primarily exert their effects by regulating epigenetic processes, serving as guides, decoys, or scaffolds to alter gene expression. Many nuclear lncRNAs function by RNA:DNA interactions or by recruiting specific proteins to their genomic targets. The list of proteins interacting with lncRNAs continues to expand, including hnRNPs, such as hnRNP-K for LincRNA-p21 [37], hnRNP-A/B and -A2/B1 for LincRNA-Cox2 [33], and hnRNPL for THRIL [36]; transcription factors like CTCF for Jpx [38], NF-kB p65 (RelA) for Lethe [34], NF-kB p50 for PACER [35]; and components of epigenetic machinery such as WDR5 for NeST and PRC2 for HOTAIR [30,39,40].

DExD/H-box RNA helicases serve as mediators of antiviral innate immunity and important host factors for viral replication. DHX9 is a multidomain member of the DExH-box superfamily 2 protein, a rich nuclear RNA helicase [41], which has been proven to participate in a variety of cellular functions, including transcription [42], translation, and innate immunity [43,44]. It is known that DHX9 can promote the replication of various viruses, including HIV-1, as well as IAV, Hepatitis C Virus (HCV), and Foot-and-Mouth Disease Virus (FMDV) [12,45,46,47]. Research has shown that IRF1-AS (an lncRNA) activates the transcription of interferon regulatory factor 1 (IRF1) by interacting with interleukin enhancer-binding factor 3 (ILF3) and DHX9 [48]. LincRNA-PILA promotes the arginine methylation of DHX9 mediated by PRMT1, leading to an increased transcription of genes encoding transforming growth factor beta-activated kinase 1 (TAK1), an upstream activator of NF-κB signaling [49].

In this study, we utilized transcriptome sequencing (RNA-seq) to demonstrate an upregulation in the expression of LINC01615 upon influenza virus infection in A549 cells. However, we observed a decrease in LINC01615 expression in HDX9-knockdown A549 cells following the influenza virus infection, compared to the wild-type virus-infected group. CLIP-seq analysis revealed an interaction between DHX9 and LINC01615. Subsequently, when using shRNAs to knock down LINC01615, we observed that the knockdown of LINC01615 promoted influenza virus replication. In summary, LINC01615 might interact with DHX9, enhancing DHX9 activity and promoting influenza virus replication.

## 2. Results

### 2.1. Intracellular LincRNA Profiles after Influenza Virus Infection

The transcriptome sequencing included a total of six samples, yielding 41.78 G of CleanData. The effective data volume for each sample ranged from 6.69 to 7.2 G, with Q30 bases ranging from 94.51% to 95.2% and an average GC content of 50.51%. By aligning reads to the reference genome, the alignment rates for each sample ranged from 97.72% to 98.31%. Principal component analysis (PCA) of the sequencing samples revealed minimal variability among the three replicates within each group, while significant differences were observed between the infection group and the MOCK group (Figure 1A). Similarly, the cluster analysis indicated good similarity among the three samples within each group (Figure 1B). Subsequently, we performed a statistical analysis of differentially expressed genes (q-value < 0.05 and |log2FC| > 1.0), revealing 336 upregulated genes and 124 downregulated genes (Figure 1C). Among these altered genes, 17 LincRNAs exhibited significant changes (q-value < 0.05 and |log2FC| > 2.0) (Figure 1D). These results suggest that LincRNAs might be involved in the viral infection processes.

### 2.2. Influenza Virus Infection Activated the Gene Expressions of Immune-Related Molecular

Upon infecting A549 cells with PR8 at an MOI of 0.05, the NP and NS1 proteins were increased with an extension of infection time, suggesting the normal proliferation of PR8 in A549 cells (Figure 2A). The expression levels of the 17 identified LincRNAs showed significant changes with varying magnitudes (Figure 2B), which were consistent with the transcriptome expression profile results (Figure 1D). Furthermore, gene set enrichment analysis (GSEA) results also indicate significant gene changes after influenza virus infection in A549 cells (Figure 2C). Additionally, the significantly changed genes enriched in the influenza A pathway were listed in the heatmap analysis (Figure 2D), among which the genes with notable changes and classical immune factors were selected for validation. The results show that the expression of these genes increased, suggesting the activation of the immune system in A549 cells after influenza virus infection (Figure 2E).

### 2.3. Knocking down Linc01615 Can Promote the Replication of the Influenza Virus

Through Integrative Genomics Viewer (IGV) analysis and shRNA screening, we focused our research on Linc01615. The interactions between LINC01615 and proteins were predicted using the catRAPID online database, and the results are shown on the following website: http://crg-webservice.s3.amazonaws.com/submissions/2023-12/775011/output/index.html?unlock=1cc5063c4c (accessed on 23 December 2023). The results indicate that there might be a potential interaction between DHX9 and LINC01615. Then, the distribution of Linc01615 in the cytoplasm and nucleus of A549 cells infected with the virus were validated. The results showed that Linc01615 was predominantly located in the cell nucleus (Figure 3A). To construct an appropriate method for knocking down Linc01615, a pair of specific shRNAs was designed to knock down Linc01615 through shRNAs. The results show that shRNAs could effectively knock down LINC01615 (Figure 3B). Furthermore, in knockdown cells infected with the PR8 virus at MOI 0.01, the results of WB showed that the expressions of NP and NS1 proteins were increased at 6, 8, and 12 h after viral infection, and protein expression levels were increased following the extended infection time (Figure 3C). These results suggest that the Linc01615 might inhibit viral proliferation in A549 cells.

### 2.4. Interaction between DHX9 and Linc01615

To investigate the relationship between DHX9 and Linc01615, we first constructed stable overexpression of DHX9 in HEK293 cells. Subsequently, we conducted cross-linking immunoprecipitation, followed by sequencing (CLIP-seq) in this cell line. After sequencing, the results were aligned with the human genome. Upon the analysis of RNA types in the immunoprecipitation (IP) samples, we found that intergenic RNA accounted for 41.73% of all the RNA components (Figure 4A). Also, the visual analysis showed that the RNA samples in the DHX9-IP group were homologous to that of the Linc01615 RNA, while the homologous RNA samples in the IgG-IP group were low, suggesting that the Linc01615 RNA bound to the DHX9 protein (Figure 4B). These results reveal a potential interaction between DHX9 and LINC01615.

### 2.5. Expression Profiles and Function Analysis of Differential Genes in Cells Expressing DHX9

After determining the RNA types in the DHX9-IP and NP-IP groups (Figure 4A), the expression profiles for the RNA from both groups were obtained, which are presented as a Venn analysis. It was found that the shared genes between the two groups accounted for 2.661% of the total, while the unique genes in the NP group accounted for 33.8% and those in the DHX9 group accounted for 63.45% (Figure 5A). Subsequently, the differential genes were subjected to heatmap analysis to classify and visually illustrate the differences of gene expression profiles between the two groups (Figure 5B). Upon identifying the differential genes, KEGG and GO analyses were performed to analyze the bioinformatics function. In the KEGG analysis results, the enriched pathways were categorized into five major classes: cellular processes, environmental information, genetic information processing, human diseases, and organismal systems. The most significant pathways identified were ubiquitin-mediated proteolysis, cell cycle, and endocytosis (Figure 5C). Additionally, in the GO analysis results, several terms related to mRNA regulation, GTPase activity, and binding were identified (Figure 5D).

### 2.6. Immune Factor Related to Interferons and ISGs upon LINC01615 Knockdown

After performing transcriptome sequencing on cells with DHX9 knockdown, followed by PR8 infection, a significantly increased number of differentially expressed LincRNAs (LINC, q-value < 0.05 and |log2FC| > 2.0) were observed. As shown in Figure 6A, only seven LincRNAs were found to be common between cells with DHX9 knockdown and wild-type cells infected with the influenza virus following the sequencing comparison, in which LINC01615 expression was decreased in A549 cells upon DHX9 knockdown after PR8 virus infection. Unexpectedly, following LINC01615 knockdown, the expression of six interferon-stimulated genes (ISGs), including IFN-β, IL-28A, IL-29, ISG-15, MX1, and MX2, were decreased, compared to that of wild-type cells (Figure 6B).

## 3. Discussion

The functional mechanism of host resistance against the influenza virus has become a topic of high concern. During influenza virus infection, the viral replication process, in which ncRNAs might interact with functional cellular proteins within the host to promote or antagonize viral replication, is complex. Non-coding RNAs (ncRNAs) play the crucial roles during cellular biological processes, whose dysregulation is often associated with diseases [50]. The function of lncRNAs is closely related to their cellular location [51]. LncRNAs in the cytoplasm can act as inducers of miRNAs or bind to ribosomes and be translated into small peptides [52,53]. Through the separation of the cell nucleus and cytoplasm and real-time fluorescence quantitative PCR, it has been determined that Linc01615 might be mostly located in the nucleus of cells [51], in which the mechanism behind the effect of LINC01615 on viral replication needs further exploration.

Here, when A549 cells were infected with PR8, 17 LincRNAs were observed to be significantly changed. The expression of Linc01615 was increased in A549 cells, and Linc01615 knockdown promoted viral proliferation and inhibited the expression of intracellular immune-related genes. On the other hand, research on LINC01615 has predominantly focused on several cancer diseases, such as colorectal cancer [50], clear cell renal cell carcinoma [51], and hepatocellular carcinoma [52]. At present, it is known that Linc-AhRA is related to the congenital antiviral response caused by the herpes neurophilus virus [53]. Linc-Pint is related to the hepatitis C virus evasion interferon [54,55], and Linc-GALMD3 may be a key regulator of Marek’s disease virus [56,57]. Various LincRNAs have been implicated in various biological processes, and each LincRNA potentially serves different intracellular functions [58]. It has been reported that Linc-satb1 mediates immune responses, while LincRNA-MSTRG.6754.1 can regulate host immunity and modulate the Wnt signaling pathway [56,57,59]. In our experiments, in the normal infection group, we observed the activation of type I and III interferons and the activation of ISGs. However, when Linc01615 was knocked down, we found that the expression of IFN-β and type III interferon were decreased. Also, ISGs were significantly reduced in Linc01615 knockdown compared to wild-type cells. These results suggest that Linc01615 might be involved in immune activation upon influenza virus infection.

Furthermore, we found that high-throughput sequencing and CLIP-seq analysis revealed the interaction between DHX9 and LINC01615. DHX9, as a multifunctional nuclear protein, has been shown to play a role in the transcription process [60]. Although DHX9 is abundant in the nucleus and a small amount of DHX9 is sufficient to maintain normal cellular function, attempts to knock out various cell lines using CRISPR/Cas9 technology have failed [61], and the homozygous deletion of DHX9 in mice can lead to embryo death [62]. As the vital protein, the functional mechanisms between nuclear DHX9 and Linc01615 are not well understood. In this study, it was validated that although upregulated in response to influenza virus infection in A549 cells, the expression of LINC01615 was reduced in A549 cells with DHX9 knockdown.

Based on a comprehensive analysis of DHX9 protein functions, we hypothesize the following scenario: when cells are stimulated by a viral infection, DHX9 is involved in the transcription process of LINC01615, leading to the formation of RNA. However, upon DHX9 knockdown, the sharp reduction in DHX9 proteins results in a decrease in the proteins available for participating in the transcription of LINC01615, which may lead to a reduction in LINC01615 levels. The exact reason why influenza virus replication is inhibited after Linc01615 knockdown, namely, DHX9, may be because Linc01615 partakes in influenza virus replication and Linc01615 helps to activate the intracellular immune system. However, the precise functional role of Linc01615 on viral replication still needs further clarification. These findings suggest a deeper connection between DHX9 and Linc01615, which highlights the significant role of Linc01615 in the influenza virus replication process. This research provides valuable insights into understanding influenza virus replication and offers new targets for preventing influenza virus infections.

## 4. Materials and Methods

### 4.1. Cell Culture and Virus

A549 cells, MDCK cells, and HEK293T cells were cultured in Dulbecco Modified Eagle Medium (DMEM) (C3103-0500, Vivacell, Shanghai, China) supplemented with 10% fetal bovine serum (FBS) (FSD500, ExCell Bio, Shanghai, China) at 37 °C and 5% CO_2_. IAV(A/PR/8/34) was amplified in 9-day-old specific pathogen-free (SPF) embryonated eggs and stored at −80 °C.

### 4.2. RNA-Seq and Data Analysis

The total RNA samples were collected from A549 cells with or without PR8 infection, and RNA-seq library preparation and sequencing for PR8-infected A549 samples were completed by OE Biotech on the Illumina Novaseq^TM^ 6000 platform (Shanghai, China). Simply, the total RNA samples were collected from A549 cells with or without PR8 infection, and RNA-seq library preparation and sequencing for PR8-infected A549 samples were completed by OE Biotech on the Illumina Novaseq^TM^ 6000 platform. Simply, the DESeq2 software, version 1.22.2 (https://bioconductor.org/packages/release/bioc/html/DESeq2.html, accessed on 23 December 2023) was utilized to normalize the counts of the genes across all the samples, and the BaseMean value was employed to estimate expression levels. Differential expression was calculated, and negative binomial distribution (NB) was employed for significance testing. The final selection of differentially expressed protein-coding genes was based on both fold change and the results of significance testing. For non-biological replicate samples, DESeq was used for the differential analysis.

### 4.3. Transfection

According to the instructions of the Lipofectamine 2000 (11668027, Thermo Fisher Scientific, Waltham, CA, USA), A549 cells were transfected with siRNAs. After 48 h, the cells were infected with the influenza virus to detect the antiviral factor expression and viral replication. The siRNA and negative control sequences that were used in this study are shown in Table 1.

### 4.4. Quantitative Real-Time PCR

The total RNA samples from the transfected A549 cells were extracted by RNAiso Plus reagent (9109, Takara, Beijing, China). The expression of antiviral factors was measured with the PrimeScript™ RT reagent Kit (RR047A, Takara, Beijing, China) and SYBR^®^ Premix Ex Taq™ (RR420A, Takara, Beijing, China). The primers of the antiviral factors are listed in Table 2. The data were normalized to GAPDH levels and further analyzed by the 2^−ΔΔCT^ method.

### 4.5. Western Blotting

The cells were washed three times with cold PBS and then collected using a lysis buffer mixture consisting of RIPA lysis buffer (R0020, Solarbio, Beijing, China) and phenylmethylsulfonyl fluoride (P0100, Solarbio, Beijing, China). The cell lysates were clarified by centrifugation at 12,000× *g* for 10 min. The protein samples were separated by 12.5% SDS-PAGE gel and then transferred onto a polyvinylidene difluoride (PVDF) membrane for antibody probing. First, the PVDF membrane was blocked with 5% skim milk. After blocking, the membrane was incubated with primary antibodies overnight at 4 °C, followed by incubation with the corresponding horseradish peroxidase-conjugated (HRP) secondary antibodies for 2 h at room temperature. Finally, the chemiluminescent substrate and an imaging system were used for membrane imaging. The primary antibodies used in this experiment included the influenza virus NP and NS1 antibodies, as well as the glyceraldehyde-3-phosphate dehydrogenase antibody.

### 4.6. CLIP-Seq

HEK293 cells stably overexpressing DHX9/NP (pEnCMV-DHX9-3×FLAG-SV40-Neo plasmid obtained from Miaoling Biotechnology, Wuhan, China) were evenly seeded in 15-cm-diameter culture dishes, with 3 dishes per sample. When the cell density reached approximately 80%, the culture medium was removed, and the cells were rinsed three times with sterile PBS buffer. The cells were then infected with the PR8 virus at a multiplicity of infection (MOI) of 0.5 for 2 h. After incubation, the virus-containing medium was discarded, and the cells were replenished with 2% serum-free DMEM in a 5% CO2 cell culture incubator for continued cultivation. After 12 h, the maintenance medium was removed, and the cells were washed three times with sterile PBS buffer. Following PBS removal, the cells were cross-linked with 1% formaldehyde at room temperature for 10–15 min. After cross-linking, formaldehyde was discarded, and cross-linking was terminated with glycine solution. The cells were then washed three times with sterile PBS buffer, collected using a cell scraper, and transferred to sterile EP tubes. The tubes were centrifuged at 1000× *g* for 5 min at 4 °C. The cells were lysed with PIPA buffer (R0278, Sigma, Shanghai, China), gently mixed by pipetting, and supplemented with protein inhibitors and RNAse inhibitors. Cell lysis was performed on ice for 10 min, during which samples were passed through a 5-mL syringe needle 3–5 times. An appropriate amount of Protein A/G magnetic beads (P2108, Beyotime, Shanghai, China) was placed into new sterile EP tubes. The tubes were placed on a magnetic rack for 2–3 min to remove the buffer, followed by the washing of the beads with RIPA buffer three times. The cell lysate was mixed with the beads and incubated at 4 °C with rotation for 1 h. After incubation, the beads were separated using a magnetic rack, and the supernatant was divided into two parts, with one part used as an IgG control and the other incubated with the corresponding antibody. An aliquot of the cell lysate was saved as the RNA input sample and stored in liquid nitrogen. Subsequently, 5 μg of antibody/IgG was mixed with the cell lysate and incubated at 4 °C with rotation for 3–5 h or overnight. After incubation, the beads were washed three times with RIPA buffer. The washed beads were resuspended in the cell lysate–antibody mixture and incubated at 4 °C with rotation for 3 h. The beads were then separated on a magnetic rack, and the supernatant was discarded. The beads were washed three times with Wash buffer. Finally, RNA was eluted from the beads by incubating at 95 °C for 2 min. The eluate was transferred to new EP tubes, and proteinase K was added to each tube. The tubes were then incubated in a 52–55 °C water bath for 1 h. The eluate was stored in liquid nitrogen for subsequent sequencing analysis. This project involved sequencing RNA samples using the Illumina PE150 platform (RiboBio, Guangzhou, China). After sequencing, the data underwent CLIP-seq bioinformatics analysis, which included quality control, sequence alignment, identification of binding peaks, motif prediction and annotation, annotation and statistical analysis of peaks, analysis of differential peaks, and KEGG and GO analyses of genes associated with differential peaks. This comprehensive analysis aimed to uncover RNA-protein interactions and associated functional pathways, providing valuable insights into the molecular mechanisms.

### 4.7. Statistical Analysis

The data were analyzed with one-way ANOVA using GraphPad Prism 6.01 statistical software, and Duncan’s post-hoc test was used for multiple comparisons. The results were expressed as mean ± standard deviation error. * *p* < 0.05; ** *p* < 0.01; and *** *p* < 0.001.

## Figures and Tables

**Figure 1 ijms-25-06584-f001:**
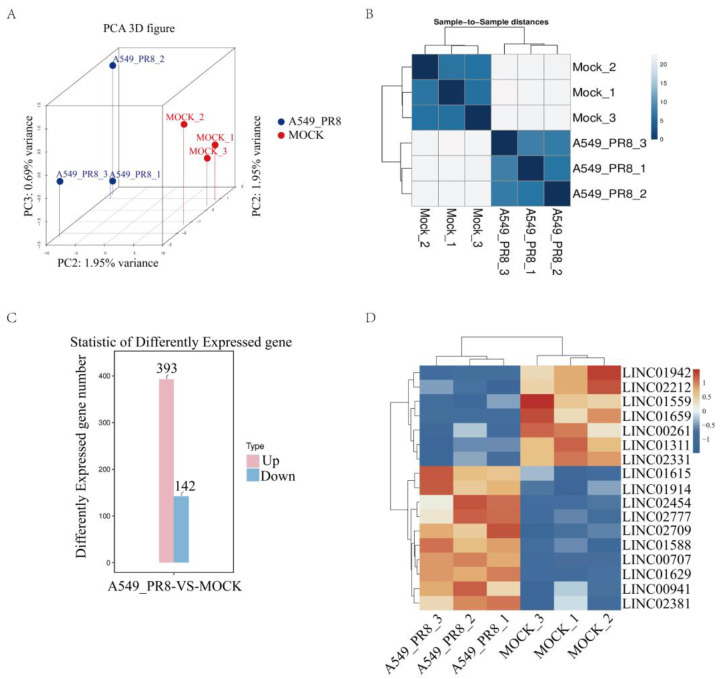
The transcriptome sequencing analysis. A549 cells were infected with or without the PR8 virus, and the total RNA from the cells was collected to be analyzed. (**A**) Principal component analysis (PCA) results. (**B**) Cluster analysis results. The horizontal and vertical coordinates are the sample name, and the color represents the size of the correlation coefficient. The horizontal and vertical coordinates represent the sample name, and the color represents the size of the correlation coefficient. (**C**) Statistical summary of differentially expressed genes. The comparison groups are on the horizontal axis, and the number of differential genes in the comparison group is on the vertical axis. Up stands for upregulated genes, and down stands for downregulated genes. (**D**) The heat map analysis of the significantly changed LincRNAs. The significantly changed LincRNAs was based on *p*-value < 0.05 and |log2FC| > 2.0. In the figure, red indicates relatively high expression protein-coding genes, and blue indicates relatively low expression protein-coding genes.

**Figure 2 ijms-25-06584-f002:**
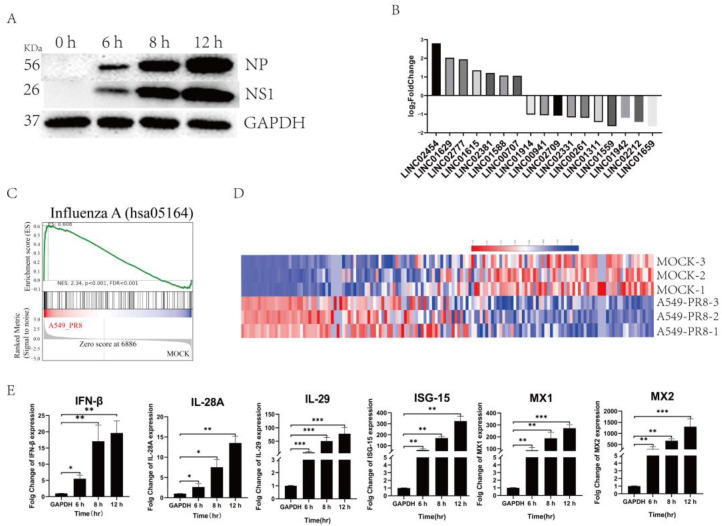
Changes in immune factors in A549 cells infected with the influenza virus. (**A**) The viral protein expression in the infected cells. A549 cells were infected with the PR8 virus, and the cells were collected at different time points to check the viral proliferation used for Western blot analysis. (**B**) The significant changes in LincRNAs based on the transcriptome sequencing results. (**C**) Gene set enrichment analysis (GSEA) in A549 cells infected with the PR8 virus. Image Description: The graphic represents the analysis result of a gene set through GSEA analysis. The graphic is mainly divided into four parts, as shown in Figure 1. From top to bottom, four aspects of the content were included, respectively. 1. Distribution plot of enrichment score (ES): The green line represents the distribution of ES for all genes. The peak position on the y-axis corresponds to the enrichment score of the gene set. 2. The vertical lines indicate the position of genes within the gene set across the entire ranking. 3. The colorbar represents the color mapping of the ranking matrix. Positive values correspond are shown in red, with larger values being darker red. Conversely, negative values are shown in blue, with larger values being darker blue. Values closer to 0 are closer to white. 4. Distribution plot of the ranking matrix: This represents the distribution of metrics such as fold change, signal 2 noise, etc. (**D**) The heat map of the enriched gene set. In the figure, red indicates a relatively high-expression coding gene, and blue indicates a relatively low-expression coding gene. (**E**) The expression of interferons and ISGs after PR8 infection. The expression changes in IFN-β, IL-28A, IL-29, ISG-15, MX1, and MX2 in A549 cells after infection with PR8 at different time points were detected with qPCR. Data are presented as the mean ± SD based on three independent experiments (* *p* < 0.05; ** *p* < 0.01, and *** *p* < 0.001).

**Figure 3 ijms-25-06584-f003:**
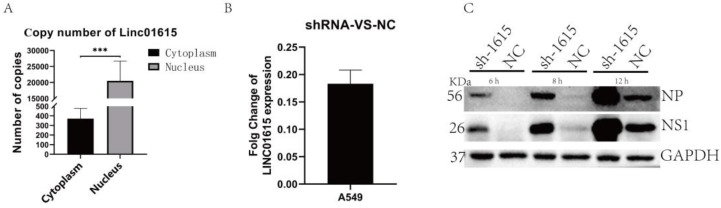
Knocking down Linc01615 promoted influenza virus proliferation. (**A**) Distribution of Linc01615 in A549 cells. After being infected with PR8 for 12 h, the nucleus and cytoplasm samples in A540 cells were separated, and the content of Linc01615 in the nucleus and cytoplasm was determined with qPCR. (**B**) The knocking down of Linc01615 with shRNAs was verified by qPCR. (**C**) The proliferation of the influenza virus in Linc01615 knockdown cells. After Linc01615 knockdown, A549 cells were infected with PR8, and protein samples were collected at 6, 8, and 12 h to analyze the expressions of NP and NS1 proteins with Western blotting. Data are presented as the mean ± SD based on three independent experiments (*** *p* < 0.001).

**Figure 4 ijms-25-06584-f004:**
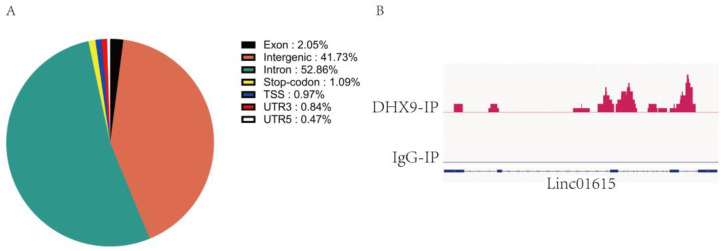
DHX9 Interaction with Linc01615. (**A**) RNA type statistics results. Cross-linking immunoprecipitation sequencing (CLIP-seq) was performed in HEK293 cell lines stably expressing DHX9. Various enriched RNA gene categories were assessed following CLIP-seq. (**B**) Integrative Genomics Viewer (IGV) analysis results. Using the human genome (GRCH38) as the reference genome, we analyzed the DHX9-IP sequencing results and visualized them against the human genome. Similarly, we performed the same operations on the sequencing results of the IgG-IP group. The absence of peaks indicates that the gene was not enriched at that location. The red peak is the homologous RNA to Linc01615 in DHX9-IP. The following is the indication area of the Linc01615 range. The larger the red area, the more homologous the RNAs.

**Figure 5 ijms-25-06584-f005:**
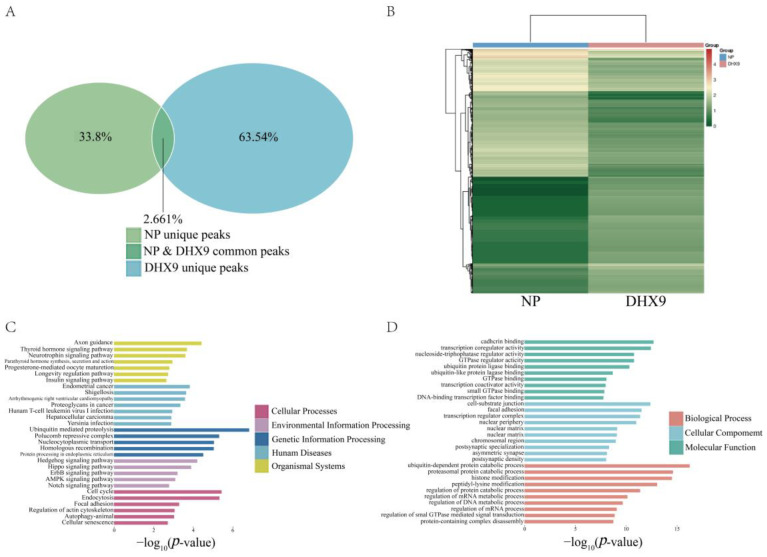
Results of differential gene analysis in DHX9 cells. (**A**) Venn analysis results of RNA in the DHX9-IP and NP-IP groups’ solutions, in which green represents the NP group and blue represents the DHX9 group. (**B**) Heatmap analysis results of differential genes between the DHX9-IP and NP-IP groups. (**C**) KEGG analysis results of differential genes. 
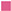
: Cellular Processes, 
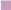
: Environmental Information Processing, 
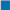
: Genetic Information Processing, 
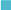
: Human Diseases, and 
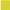
: Organismal Systems. (**D**) GO analysis results of differential genes, 

: Biological Processes, 
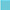
: Cellular Components, and 
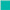
: Molecular Functions.

**Figure 6 ijms-25-06584-f006:**
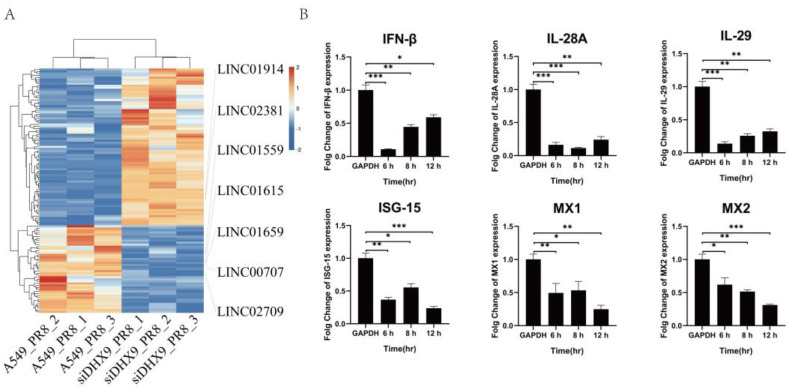
The expression of interferons and ISGs after PR8 infection. (**A**) The heat map analysis of the significantly changed LincRNAs after DHX9 knockdown. The significantly changed standard of LincRNAs is as follows: q-value < 0.05 and |log2FC| > 2.0. In the figure, red indicates a relatively high-expression coding gene, and blue indicates a relatively low-expression coding gene. (**B**) Changes in six interferons and ISGs in Linc01615 knockdown A549 cells infected with the PR8 virus. The cells in the experimental group were treated with the knockdown of Linc0615, and the cells in the control group were treated with the corresponding empty vector. The expression changes in IFN-β, IL-28A, IL-29, ISG-15, MX1, and MX2 in two groups of A549 cells after infection with PR8 at different time points were detected with RT-qPCR. Data are presented as the mean ± SD based on three independent experiments (* *p* < 0.05; ** *p* < 0.01, and *** *p* < 0.001).

**Table 1 ijms-25-06584-t001:** Primer sequences of the small interfering RNA and the negative control group used in this study.

Primer	Sequence
siDHX9-as	5′-AUCAAUGUUGCUACUAGUCTT-3′
siDHX9-ss	5′-GACUAGUAGCAACAUUGAUTT-3′
siNC-as	5′-UUCUCCGAACGUGUCACGUTT-3′
siNC-ss	5′-ACGUGACACGUUCGGAGAATT-3′

Note: as: sense sequence; ss: antisense sequence; and NC: Negative control.

**Table 2 ijms-25-06584-t002:** The specific primer sequences used in this study.

Primer	Sequence
IFN-β-sense	5′-AATTGCTCTCCTGTTGTGCTTCTCC-3′
IFN-β-antisense	5′-GTCAATGCGGCGTCCTCCTTC-3′
IL-28A-sense	5′-GCCCTGACGCTGAAGGTTCTG-3′
IL-28A-antisense	5′-GCGGAAGAGGTTGAAGGTGACAG-3′
IL-29-sense	5′-CCGTGGTGCTGGTGACTTTGG-3′
IL-29-antisense	5′-GTTGTGGTGGGCTTGGAAGTGG-3′
ISG-15-sense	5′- AATGCGACGAACCTCTGAACATCC-3′
ISG-15-antisense	5′-CGAAGGTCAGCCAGAACAGGTC-3′
MX1-sense	5′-CTCCGACACGAGTTCCACAA-3′
MX1-antisense	5′-GGCTCTTCCAGTGCCTTGAT-3′
MX2-sense	5′-AGGTTCCAGACCTGACCATC-3′
MX2-antisense	5′-GTCTGCTGCCTCTGGATGTA-3′
GAPDH-sense	5′-GAGTCAACGGATTTGGTCGT-3′
GAPDH-antisense	5′-GACAAGCTTCCCGTTCTCAG-3′

## Data Availability

The data presented in this study are available upon request from the corresponding author.

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
