# Peer review of "The Identification and Function of Linc01615 on Influenza Virus Infection and Antiviral Response"

_ijms, 2024, doi:10.3390/ijms25126584_

Round 1
Reviewer 1 Report
Comments and Suggestions for Authors
In this work, the authors have explored the role of Linc01615 long non-coding RNA in influenza virus replication using various molecular biology techniques. However, the authors did not adequately explain their results, which makes it difficult to understand why they used some experiments to identify the role of this non-coding RNA. There are several missing links in the study.
- Line 155- The expression levels of the identified 17 LincRNAs showed significant changes with varying magnitudes (Fig. 2B). However, figure 2B shows less than 17 LincRNAs. Could authors explain why few LincRNAs were omitted?
- In section 2.3, the authors jump directly into Linc01615 RNA. The authors must explain to the readers why Linc01615 was selected for downstream studies among 17 RNAs.
- Line 193- a pair of specific shRNA 193 were designed to knock down Linc01615. The authors claim that Linc01615 is predominantly present in the nucleus region. If the ncRNA is nuclear, while the shRNA machinery is cytoplasmic, knockdown will be more difficult. On top of this, in line 15, the authors say that the Crispr-Cas9 system was used to knock down Linc01615, not shRNA.
- In Figure 3C, the authors showed that when Linc01615 was knocked down in A549 cells, the expression of NS1 and NP proteins was significantly high compared to non-knockdown cells. However, in Figure 1A, the authors showed high expression of NS1 and NP proteins in wild-type cells in the same time frame, which is contradictory.
- Why was NP-IP performed?
- Linc01615 was knockdown promoting or suppressing viral replication? In line 16, virus proliferation is suppressed, but in 19, virus replication is promoted.
- Figure legends are not legible.
- How was DHX9 knockdown performed?
Author Response
Dear Editor and Reviewers,
Thank you for taking the time to review our manuscript and for providing valuable feedback and suggestions. We greatly appreciate your efforts and have carefully revised our manuscript in response to your comments. Below, we provide detailed responses to each of the reviewers' comments.
- In Line 155 in the revised manuscript, the data about LincRNAs in results of Fig.2B is checked and revised. The discrepancy in LincRNAs quantity was due to an inadvertent omission of some LincRNAs in Fig.2B. The thorough re-analysis of the data is conducted, including all 17 LincRNAs as initially identified. The revised Fig.2B in the revised manuscript accurately reflects the expression levels of all the LincRNAs under study.
- In section 2.3 in the revised manuscript, selection of Linc01615 for downstream studies was based on the following aspects: (1) Visualization analysis of DHX9 and IgG sequencing results using IGV software; (2) Screening of LincRNAs using shRNA that impacts influenza A virus; (3) Hypothesized potential roles of LincRNAs based on existing evidence. These comprehensive criteria led us to focus on Linc01615. Regarding your concern, we will include this explanation in the revised manuscript.
- In Line 193 in the revised manuscript, a pair of specific shRNA were designed to knock down Linc01615. We initially utilized the CRISPR-Cas9 system for LincRNA screening. However, in our preliminary experiments, we found that using CRISPR-Cas9 to knock down Linc01615 did not effectively demonstrate the knockdown effect and had no impact on influenza virus replication. On the other hand, we found evidence in Yi Zhang's article (DOI: 10.1007/s00018-022-04675-7) that shRNA can successfully knock down Linc01615. Thus, we opted to use shRNA for subsequent experiments. The mention of using CRISPR-Cas9 to knock down Linc01615 in the manuscript was an oversight, which is checked and revised in the revised version.
- In the revised manuscript, the data of Figure 3C showed that when Linc01615 was knocked down in A549 cells, the expression of NS1 and NP proteins was significantly high compared to non-knockdown cells. However, Figure 1A is a Principal Component Analysis (PCA) that demonstrates the reproducibility among three replicate samples within each group and the overall differences between the transcriptomes. The data of Figure 1A provides an overall assessment of each sequencing sample and does not reflect the expression levels of specific genes. Additionally, the transcriptome sequencing was performed on A549 cells, so the results only show changes in the A549 cell transcriptome and do not include the expression levels of influenza virus genes. Therefore, we believe that the results shown in Figure 1A do not conflict with those presented in Figure 3C.
- In another manuscript currently under submission, we have demonstrated that the interaction between DHX9 and the influenza virus NP protein facilitates the transcription of the influenza virus. Given that LincRNAs typically function by forming complexes with host proteins, and since we have shown the interaction between DHX9 and Linc01615, we were interested in exploring whether DHX9 and Linc01615 form a complex and whether this complex plays a role in the transcription process of the influenza virus. This is a significant research direction. Consequently, we also performed NP-IP analysis to provide data support for future experiments.
- In the revised manuscript, Linc01615 knockdown actually promotes viral replication. The unreasonable description has been corrected in the revised manuscript.
- In the revised manuscript, Figure legends are revised.
- In the revised manuscript, we utilized siRNA to knock down the DHX9 protein. Initially, we attempted to use the CRISPR-Cas9 system to knock out DHX9, but these efforts ultimately proved unsuccessful. Upon further investigation, we discovered that attempts to knock out the DHX9 gene using the CRISPR/Cas9 system in various cell lines typically resulted in failure. Additionally, homozygous deletion of DHX9 in mice leads to embryonic lethality. Therefore, we opted for the more practical approach of using siRNA to knock down the DHX9 protein.
Thank you again for your valuable feedback and constructive comments. We believe that the revisions have significantly improved the quality of our manuscript. We look forward to your positive response.
Best regards,
Guihu Yin
First Author
Reviewer 2 Report
Comments and Suggestions for Authors
Title: The identification and function of Linc01615 on influenza virus infection and antiviral response
In this paper the authors study the influenza virus in its replication in relation to the immune system. The authors conclude that this article provides valuable insights into understanding influenza virus replication and offers new targets for preventing influenza virus infections.
For greater understanding of the article, the genetic steps with acronyms should be explained.
Figures 4 and 5 should be better explained.
Table 1 and 2 present only the title without any explanation.
Since this paper talks about viral replication, to be more up to date, it should include some references to SARS-CoV-2. Therefore, to make this article more interesting for the readers of this important journal, the authors should expand a bit the discussion. Here, 3 articles has been recently reported that should be studied, incorporate the meaning and report briefly in the discussion and in the list of references.
G. Anogeianakis. COVID-19: THE OMICRON B.1.1.529 VARIANT. European Journal of Neurodegenerative Diseases September-December 2023; 12(3):91-93 (www.biolife-publisher.it).
E. Antoniades, S. Melissaris, D. Panagopoulos, E. Kalloniati, G. Sfakianos. Pathophysiology and neuroinflammation in COVID-19. European Journal of Neurodegenerative Diseases 2022; 11(1) January-June: 7-9. (www.biolife-publisher.it).
S.K. Kritas. COVID-19 and pain. European Journal of Neurodegenerative Diseases 2021; 10(2) July-December: 32-35. (www.biolife-publisher.it).
I believe these suggestions are important for improving this paper. Without these corrections the paper cannot be published. So I recommend minor revision.
Comments on the Quality of English LanguageMinor editing of English language required
Author Response
Dear Editor and Reviewers,
Thank you for taking the time to review our manuscript and for providing valuable feedback and suggestions. We greatly appreciate your efforts and have carefully revised our manuscript in response to your comments. Below, we provide detailed responses to each of the reviewers' comments.
- In the revised manuscript, to enhance clarity and facilitate reader understanding, we supplement the detailed explanations of the genetic steps along with their corresponding acronyms throughout the manuscript.
- In the revised manuscript, Figures 4 and 5 are more thoroughly explained to aid in comprehension. We provide the clearer legends and potentially add annotations to the figures to elucidate key points.
- In the revised manuscript, the data of Tables 1 and 2 are revised to include explanations alongside the titles, which provide the context for readers to better understand the data presented.
- In the revised manuscript, we acknowledge the importance of discussing relevant research on SARS-CoV-2 in our manuscript. We will expand the discussion section to incorporate recent findings from three pertinent articles related to viral replication. Additionally, we will include these references in the list of citations to provide readers with comprehensive and up-to-date information on the topic.
Thank you again for your valuable feedback and constructive comments. We believe that the revisions have significantly improved the quality of our manuscript. We look forward to your positive response.
Best regards,
Guihu Yin
First Author
Reviewer 3 Report
Comments and Suggestions for Authors
The authors found increased expression of Linc01615 in A549 cells after influenza virus infection, which influenced the initiation of the study. Using the Crispr-Cas9 system to knock down Linc01615, it was found that virus proliferation was suppressed and the expression of IFN-b, IL28, IL29, ISG15, MX1 and MX2 was inhibited. After further research, the authors demonstrated an association between DHX9 and Linc01615, which highlights the significant role of Linc01615 in the influenza virus replication process. The research was well designed and the methodology was well described in detail.
Unfortunately, the figures are small and of poor quality, which means that little is known about them.
Comments on the Quality of English LanguageOK
Author Response
Dear Editor and Reviewers,
Thank you for taking the time to review our manuscript and for providing valuable feedback and suggestions. We greatly appreciate your efforts and have carefully revised our manuscript in response to your comments. Below, we provide detailed responses to each of the reviewers' comments.
In the revised manuscript, the image quality of figures are revised. To address this concern, we will ensure that the figures are resized to a larger format and improved in quality to enhance visibility and clarity. Additionally, we will provide detailed explanations and annotations to accompany the figures, facilitating better understanding of the content they represent. We appreciate your feedback and are committed to ensuring that our figures meet the highest standards of clarity and accessibility.
Thank you again for your valuable feedback and constructive comments. We believe that the revisions have significantly improved the quality of our manuscript. We look forward to your positive response.
Best regards,
Guihu Yin
First Author
Round 2
Reviewer 1 Report
Comments and Suggestions for Authors
This is regarding comment 4.
Apologies for the typo in my previous comment. It was not Figure 1A. It is Figure 2A.
In Figure 3C, the authors showed that when Linc01615 was knocked down in A549 cells, the expression of NS1 and NP proteins was significantly high compared to non-knockdown (wild-type) cells. However, in Figure 2A, the authors showed high expression of NS1 and NP proteins in wild-type cells in the same time frame, which is contradictory. Kindly address this one.
Author Response
Response to Reviewer 1 Comments
Thank you for your comments. We understand your concern regarding the apparent discrepancy between Figures 2A and 3C. We speculate that the difference between the two results may be caused by the following reasons:
Firstly, the experimental conditions is different.
Figure 2A depicts the expression of NS1 and NP proteins in wild-type A549 cells without any treatment.
Figure 3C shows the results of an experiment where A549 cells were subjected to transfection. Our research confirms that liposome treatment has a slight impact on cellular processes and protein expression levels. The negative control (NC) group in Figure 3C was also treated with liposomes, which may be one of the reasons for the slightly lower viral protein expression in Figure 3C.
Secondly, the exposure time is different.
In Figure 3C, the exposure time was adjusted to accommodate the high concentration of viral proteins in the Linc01615 knockdown group. This adjustment was necessary to avoid overexposure and ensure clear visualization of the bands. Consequently, the NC group in Figure 3C may appear less prominent compared to Figure 2A.
Thirdly, there are two independent experiments.
The experiments presented in Figures 2A and 3C were conducted independently. Figure 3C focuses on the relative changes in protein levels between the knockdown and control groups. Despite the differences in represented protein levels of the target protein band, the trend observed in the NC group in Figure 3C is consistent with the wild-type results in Figure 2A.
Finally, there are some variabilities in experimental conditions.
Differences in protein levels between Figures 2A and 3C could also result from variations in experimental conditions, such as antibody incubation temperature or differences in handling by different researchers.
Although there may be some differences in the data, the results reflected in Figures 2C and 3C are consistent. We hope these explanations address your concerns. Please feel free to contact us if you have any further questions. Thank you.
Best regards,
Guihu Yin
First author
Round 3
Reviewer 1 Report
Comments and Suggestions for Authors
I am satisfied with the author's explanation, and I have no further comments.